# Quality Characteristics of Beef Patties Prepared with Octenyl-Succinylated (Osan) Starch

**DOI:** 10.3390/foods10061157

**Published:** 2021-05-21

**Authors:** Mohamed F. Eshag Osman, Abdellatif A. Mohamed, Mohammed S. Alamri, Isam Ali Mohamed Ahmed, Shahzad Hussain, Mohamed I. Ibraheem, Akram A. Qasem

**Affiliations:** Department of Food Science and Nutrition, King Saud University, Riyadh 1145, Saudi Arabia; moh.fareed77@yahoo.com (M.F.E.O.); msalamri@ksu.edu.sa (M.S.A.); iali@ksu.edu.sa (I.A.M.A.); shhussain@ksu.edu.sa (S.H.); mfadol@ksu.edu.sa (M.I.I.); aqasem@ksu.edu.sa (A.A.Q.)

**Keywords:** beef patties, corn starch, Osan, tenderness, cooking loss

## Abstract

Octenyl-succinylated corn starch (Osan) was used to improve the physicochemical properties of ground beef patties. The study involved incorporation of 5 and 15% Osan and storage for 30 or 60 days at −20 °C. The tested parameters included cooking loss, microstructure image, firmness, color, and sensory evaluation of the prepared patties. Along with Osan, native corn starch was used as control and considered the patties with added animal fat. The data showed that Osan reduced the cooking loss and dimensional shrinkage significantly (*p* < 0.05), whereas the moisture retention, firmness and color of beef patties were improved. The sensory evaluation indicated enhanced tenderness and juiciness without significant alteration of flavor, color, and overall acceptability of the cooked patties. Microstructure images of cooked patties indicated uniform/cohesive structures with small pore size of patties shaped with Osan. Obviously, good storability of the uncooked patties was reflected on the physiochemical, textural, color, and sensory evaluation of the cooked patties, which points to the benefit of using Osan in frozen patties and signifies possible use in the meat industry. The overall sensory acceptability scores were given to cooked patties containing Osan starch as well as the native starch, whereas 15% animal fat was favored too.

## 1. Introduction

Meat patties are considered the most popular ready to eat food, due to its desirable sensory and mouth feel. This qualifies its consumption to be considered as a good part of human diet in the past few decades, in addition to its nutritional value which includes essential amino acids, vitamins, and minerals [1]. In processed meat, animal fat is a principal ingredient, found in lumps and plays a central functional and sensory role such as a binding and flavoring agent [2]. However, fat content is a critical obstacle that needs to be addressed especially at high levels. Specifically, fat has been associated with chronic deceases, such as obesity and high cholesterol which leads to hypertension and cardiovascular diseases [3]. Therefore, reducing fat turned into a current trend for the meat-processing industry to meet the consumer demands [4]. Heretofore, countless efforts have been devoted to address this concern in terms of preparing healthy products without reducing the characteristics of the final full fat product such as the physical appearance and sensory qualities. Carbohydrate-based fat replacers, such as inulin, gums, cellulose derivatives, and starches are wildly considered as fat replacer and/or fat substitute applied in low fat meat products [5,6], because of its abundancy, superior functionality, and cost-effectiveness [7,8]. Among these carbohydrates, starch exhibited extremely good functional properties improving viscosity, solubility, and water-holding capacity, as well as, adaptability [9].

Because of its poor water solubility and high retrogradation, native starches utilization is limited. This presented the need for physical, chemical, or enzymatic modifications of native starches so as to expand its utilization [10]. Starch modification can improve its functionality such as viscosity, become more tolerant to various processing conditions such as extreme pH, high temperatures, and shear. For instance, chemically modified starches exhibit significant change in its functional properties compared to physical or enzyme modifications. One of the common chemically modified starches is octenyl-succinylation (Osan) which accrues by esterification of some OH groups by octenyl molecules [11]. To overcome the hydrophilic nature of the abundant hydroxyl groups, various starch modifications by introduction of hydrophobic moiety have been reported. Therefore, amphiphilic modification of starch is one of the methods widely used to improve their hydrophobicity, because amphiphilic starches have a wide range of applications, mainly in emulsification and encapsulation. The amphiphilicity of octenyl succinic anhydride (OSA)- modified starch is improved due to the introduction of dual functional hydrophilic and hydrophobic groups [12]. The obtained Osan starch was utilized in various application ranging from pharmaceutical to food products, such as, puddings, sauces, and baby foods [13]. The use of sodium octenyl succinate starch in a methacrylate/polysaccharides blends, introduced good flowability, surface-active, smoother particle surface, and low-viscosity to spray dried emulsion of the blends [14]. In addition, Osan starch has been used as a fat substitute because it enhances the firmness and high palatability of some meat products [15]. In contrast, octenyl-modified waxy maize starch was used successfully in low fat mayonnaise at substitution level up to 75%, where the product exhibited great sensory quality such as texture and aroma [13]. In baked products, Osan starch improved dough machinability and handling as well as the loaf volume [16].

The objective of this work is to compare the quality characteristics of beef patties prepared with starch to those with animal fat. Therefore, ocetinyle succenylated and native corn starch as fat replacement in beef patties was utilized. This work includes the effect of storability at −20 °C on the quality characteristics of the patties (microstructure, cooking properties, moisture retention, texture, color, and sensory characteristics of the produced patties).

## 2. Materials and Methods

### 2.1. Materials

Lean beef meat and animal fat (Beef) were purchased from a central meat market (Al-Taamir Market, Riyadh, Saudi Arabia). Ground lean meats were selected from animals of the same age, breed, and feeding protocol (aldanon farm). While, fresh fat was obtained from Riyadh slaughterhouse (Riyadh, Saudi Arabia). Corn starch was purchased from Middle East Food Solutions Company (Riyadh, Saudi Arabia). Black paper, white paper, onion powder, garlic powder, were purchased from Panda Retail Company (Riyadh, Saudi Arabia). Analytical grade reagents, HCL, 2-Octen-1-yl succinic anhydride, NaOH were purchased from Sigma-Aldrich Chemical Co. (St. Louis, MO, USA).

### 2.2. Methods

#### 2.2.1. Octenyl-Succinylation of Corn Starch (Osan)

Osan corn starch was synthesized via esterification according to the method of [17] with slight modification. Starch, (500 g) was suspended in 1125 mL of distill water. The pH of the slurry was adjusted to 8.5–9 using 1% NAOH, followed by the addition of Osan (4% based on starch dry weight). The 2-Octen-1-yl succinic anhydride was added slowly with continuous agitation at 35 °C, while maintaining the pH around 8.5–9.0 the reaction was allowed to continue for one hour and the final pH was adjusted to 6.5 using 1.0 N HCL. The obtained mixture was centrifuged at (4000× *g*) for 10 min, washed twice with distill water and once with acetone. The product was dried at room temperature for two days, ground, sieved through 250 µm sieve and stored for further use. This product is considered as food grade according to the followed preparation method.

#### 2.2.2. Fourier Transform Infrared Spectroscopy (FT-IR)

The stretching vibrational mode of the functional group (Osan) on the modified starch was detected by FTIR spectrophotometer (Bruker, ALPHA, Hanau Germany). One gram of dry sample was placed on the FT-IR cell and scanned between 4000 and 500 cm^−1^.

#### 2.2.3. Preparation and Processing of Beef Patties

Beef patties were prepared and processed as described by Alejandre et al. [18] with slight modification. Round beef cut was sliced to small pieces and the intramuscular fat was removed. The obtained lean meat was ground to passed through a 5-mm plate meat grinder EMG-1600R ELEKTA ltd, Elekta (Hong Kong, China). Simultaneously, the beef caul fat was melted for 5 min in the microwave oven (sharp, 1200 W output, (Osaka, Japan). Six formulations with the following common ingredients g/100 g addition based on ground lean meat weight (100 g), where 1.0 g of the following ingredients were added salt, hot pepper, white pepper, dry powdered garlic, dry onion powder, and 0.35 g of vinegar, in addition to 20 g of ice water. The experimental design included two treatments and three subsamples; two levels of caul fat, two levels of native corn starch, and two levels of Osan corn starch, hence, the one level included 5% and the second was 15%. Samples with caul fat are considered the control and the melted caul fat was added to the spiced lean ground beef drop-wise with constant hand mixing, but in the starch-containing formulation the caul fat was totally replaced by modified or native starch which was added in small amounts while mixing using a Stephan UM 12 mixer (Stephan U. Sohner GmbH and Co., Gackenbach, Germany). Patties (100 g) were prepared using a patty-making machine (Expro. Co., Shanghai, China). The compressed patties, 100 mm diameter and 10 mm thick, were packaged in vacuumed plastic bags and stored at −20 °C for 0, 30, and 60 days. Before analysis, frozen un-cooked patties were placed at room temperature for 1.5 h. The raw patties were cooked in two steps; the first step was to prepare precooked patties by steaming for 20 min to stabilize the diameter, whereas the second step was the final cooking using the electric hot plate (Stilfer model, 0040, Genova, Italy) for a total of 10 min with 5 min on each side at 180 ± 1 °C. The internal temperature of the patty was 75 °C measured at the geometrical center of the patties using digital thermocouple probe (Ecoscan Temp JKT, Eutech instruments, Pte Ltd., Keppel Bay, Harbour Front, Singapore) The sensory evaluation of the cooked products was carried out directly after cooking.

#### 2.2.4. Scanning Electron Microscopy (SEM)

Microstructures of the obtained cooked patties was examined using JEOL-6360A SEM (Jeol Ltd., Tokyo, Japan). Samples were cut into small pieces 5 × 5 × 1 mm, mounted on the pin stubs using copper tape before coating with gold using an automated sputter coater JFC-1600 Auto Fine Coater (Jeol Ltd., Tokyo, Japan) for 5 min at 2.5 kV operation energy. Subsequently, four fields of each sample were spotted and the selected images were captured at magnifications ranging from 100× to 1000×.

#### 2.2.5. Measurement of Cooking Parameters

##### Cooking Loss

The cooking loss of patties was determined by weighing before and after cooking as recommended by Hollenbeck et al. [19] using the following equation:Cooking loss = ((un−cooked patties weight)−(cooked patties weight))/(un−cooked patties weight)*100

##### Moisture Content

The moisture contents of un-cooked and cooked patties were determined based on the AOAC Method 950.46 [20].

##### Moisture Retention

The moisture retention, was determined as the amount of moisture retained in the cooked product per 100 g of raw sample. This value was calculated according to the following equation described by [21,22], where the weight of the patties was recorded before and after cooking, and the cooking yield was calculated by dividing the weight of cooked patties by the weight of uncooked patties and expressed in as reported by [23].
$$\mathrm{Moisture}\;\mathrm{retention}\;=\;\mathrm{cooking}\;\mathrm{yield}\;\times \frac{\mathrm{moisture}\;\mathrm{in}\;\mathrm{coocked}\;\mathrm{patties}}{\mathrm{moisture}\;\mathrm{in}\;\mathrm{un-cooked}\;\mathrm{patties}}$$
Cooking yield = weight before cooking − cooked weight

##### Patties Diameter, Thickness, and Shrinkage

Change in beef patties’ diameter was determined before and after cooking using Digital Electric Caliper (Pen Tools Co., Maplewood, NJ, USA) by employing the following equations.
$$\mathrm{Diameter}\;=\;\frac{\left(\mathrm{un-cooked}\;\mathrm{patties}\;\mathrm{diameter}\;\right)-\left(\mathrm{cooked}\;\mathrm{patties}\;\mathrm{diameter}\right)}{\mathrm{un-cooked}\;\mathrm{patties}\;\mathrm{diameter}}\;\times \;100$$
$$\mathrm{Thickness}\;=\;\frac{\left(\mathrm{un-cooked}\;\mathrm{patties}\;\mathrm{thickness}\;\right)-\left(\mathrm{cooked}\;\mathrm{patties}\;\mathrm{thickness}\right)}{\mathrm{un-cooked}\;\mathrm{patties}\;\mathrm{thickness}}\;\times \;100$$
$$\mathrm{Dimensional}\;\mathrm{shrinkage}\;=\;\frac{\left(\mathrm{Raw}\;\mathrm{thickness}-\mathrm{cooked}\;\mathrm{thickness}\right)+\left(\mathrm{Raw}\;\mathrm{diameter}-\;\mathrm{cooked}\;\mathrm{diameter}\right)}{\mathrm{Raw}\;\mathrm{thickness}-\;\mathrm{raw}\;\mathrm{diameter}}\;\times \;100$$

##### Firmness

The firmness of cooked beef patties was determined using a texture analyzer (TA XT Express, Micro Systems Ltd., Surrey, UK). Samples (60 mm diameter and 10 mm thickness) were pressed using aluminum cylinder probe (SMS P/20 mm diameter, TA XT Plus Micro Systems Ltd., Surrey, UK) operated at 1 mm/s. Samples were compressed to 8 mm distance with 10% strain, where the needed force is expressed in Newton (N). The shear force corresponds to maximum peak force, expressed in Newton (N). The test was performed at room temperature (25 ± 1 °C).

##### pH

The pH was measured using a portable pH-meter (Model pH 211, Hanna Instruments, Woonsocket, RI, USA) by injecting the probe in 25 g of meat patty and held for 10 s to obtain the pH value.

##### Surface Color Measurement

The surface color characteristics of un-cooked and cooked patties were determined after the specified storage time (0, 30, or 60) days. The measurements included, lightness (L*), redness (a*), and yellowness (b*), assessed using a portable colorimeter (Konica Minolta, CR-400-Japan; Measuring aperture: 8 mm; Illuminant: CIE D65; Observer angle: CIE 2° Standard Observer). Five color measurements were done, where each patty was separated into four quarters one measurement on the surface of each quarter was taken and the fifth was done in the middle.

#### 2.2.6. Sensory Evaluation of Meat Patties

The sensory test was performed using a 9 points hedonic test (Affective Tests), which includes scale from 1 (dislike extremely) to 9 (like extremely) in a single session. This test is useful for evaluating the acceptance of new products. The sensory evaluation team included trained students and King Saud University staff average age between 22 and 60 years old. The panelists were trained to be able to evaluate the sensory properties of patties including overall product acceptability according to method of [24]. After training, 13 panelists were selected based on their ability and sensitivity to point out differences between the parameters. Cooked patties were tested on a 9-points scale method. The test was conducted in a designated sensory evaluation laboratory with appropriate setting such as partitioned cabinets and individual lightning at 20 ± 2.0 °C. Six treatments with two levels of fat content (control), native or modified starch were evaluated. Patties were cooked as described above and cut into triangles (25 × 20 mm) and served warm to the participants. Water and mint were also provided to neutralize the flavor between samples [25]. The expert panelists were asked to evaluate the color, flavor, tenderness, juiciness, and overall acceptability.

#### 2.2.7. Statistical Analysis

The statistical analysis was carried out using the Tukey HSD test (Statistic 10 Data analysis software, Inc., Chicago, IL, USA) at (*p* ≤ 0.05). The significance level of the analysis of variance (ANOVA) was applied to observe the differences. All measurements were done in triplicate.

## 3. Results and Discussion

### 3.1. Fourier Transform Infrared Spectroscopy (FT-IR)

FT-IR profile is shown in Figure 1. In general, the highly intense peaks noticed around 3430 cm^−1^ were ascribed to (O-H) characteristics stretching vibration of amylose or amylopectin, while peaks around 2930 and 1645 cm^−1^ are attributed to C–H stretching and to the tightly bound water present in the starch, respectively [26]. In addition, the peak at 1020 cm^−1^ was originated from the C–O stretching vibration of glucose monomer (Garcia and Grossmann 2014). Two new peaks emerged after modification. Evidently, the region between 1720 and 1570 cm^−1^ is considered a finger print for the main functional groups of the octenyl-succinylated (Osan) corn starch [27]. The new peak at 1571 cm^−1^ emerged after OSA modification was ascribed to the asymmetric stretching vibration of carboxylate RCOO–, whereas the other new peak at 1725 cm^−1^ was observed, which can be attributed to the characteristic C=O stretching vibration of an ester carbonyl group [28].

### 3.2. Scanning Electron Microscopy (SEM)

The gel network of patties with 5% addition of either fat or starch, was loose and irregular, because large holes emerged within the structure compared to the 15% (Figure 2). Nonetheless, the gel network structure of the 15% Osan starch was more compact and dense. Furthermore, smaller holes were observed in the surface of the patties with native starch and the holes were more obvious with the increase in the amount of native starch in the samples. Moreno et al. [29] reported better surface structure using a muscle homogenizer than samples with added sodium alginate as a cold gelation technology. Tseng et al. [30] reported dense SEM images of meat balls treated with TGase enzyme compared to untreated samples which indicates rise in the formation of intermolecular ε (γ-glutamyl)-lisil cross-links due to the action of the enzyme. In this study, starch generated a dense network by absorbing the excess water released by the meat during cooking which leads to swelling and closing of the gaps created during cooking. Therefore, smoother denser surface was formed in the presence of starch and more so Osan starch compared to the control. Other researchers used plant material rich in hydrofoils reported improved microstructure of meat and homogenous network [31].

### 3.3. Physical Properties of Beef Patties

#### 3.3.1. Cooking Properties

The cooking properties of the patties prepared with Osan-corn starch, native corn, and control are presented in Table 1. The addition of Osan-corn starch at 5 or 15%, significantly (*p* ≤ 0.05) altered the cooking loss, moisture retention, thickness, diameter, and dimensional shrinkage of the patties. These parameters were compared to the raw patties. The data presented here are in agreement with the literature reports on the effectivity of corn starch to retain moisture of the cooked bologna [10,32,33]. The firmness of patties prepared with native starch was significantly higher than Osan starch which can be attributed to the amylose retrogradation which is less in Osan starch, thereby, patties with more starch exhibited firmer texture (Table 1) [34]. The firmness of the control was much higher at 5% fat content compared to 15% indicating softer texture due to higher fat content but, it remains significantly higher than those with either type of starch, especially Osan starch. This could be accredited to the incapacity of fat to retain moisture during cooking compared to the starch, which is in line with the water retention property difference of the patties stated in Table 1 [35]. The pH values of the patties were stable throughout the process, therefore, the use of starch did not affect the pH of the patties during storage or after cooking. The percent diameter reduction after cooking was 18.3 for the 5% fat content (control) and 23.5 for the 15% fat content, whereas for the Osan starch patties it was 7.7 and 8.6%, while the native starch exhibited reduction as 13.4 and 16.45%, respectively. This shows significantly lower diameter reduction of Osan starch patties compared to the control and the native starch. Thereby, the stability of the patties network structure due to Osan starch swelling and the formation of a semi solid gel that stabilized the diameter, is evident (Table 1). This can be attributed to the amphiphilic (hydrophilic-hydrophobic nature) property of the Osan starch. The control exhibited the greatest diameter loss after cooking by virtue of increased animal fat content. Park et al. [36] reported diameter and thickness reduction for pork patties decreased with increase in the content of black rice powder (rice powder is about 75% starch). The dimensional shrinkage followed the same trend as the diameter reduction. Consequently, the score of the cooking loss, moisture retention, diameter reduction, and dimensional reduction favored Osan over native starch. Cornejo-Ramírez et al. [37] reported that water absorption, swelling power, and viscosity of Osan starch is superior to native starch. The high moisture retention of Osan starch lead to highly viscous gel with emulsifying power capable of holding fat and water and form a gel with little un-noticeable pores (space within the structure) and improve the sensory characteristics of the patties [33]. On the other hand, the control beef patties exhibited poor binding, limited protein network structure, or entrapments of ingredients as shown by the sizable pores in the protein network. These results are in agreement with [38,39,40,41]. The thickness of the control increased at higher fat content, but with native starch, significantly lower thickness was observed at high starch content. Osan starch patties exhibited significantly higher thickness compared to the control and the native starch, since the thickness was almost twice as much (Table 1).

#### 3.3.2. Effect of Storage on the Cooking Properties

The effect of storage on cooking loss, firmness, and other cooking properties is presented in Table 2. Although longer storage time significantly increased the tested parameter of patties containing 5% starch or animal fat, Osan starch performed better than the control or the native starch under the same storage conditions. The storability of patties significantly enhanced when 15% of starch or animal fat were added. Osan starch improved all tested parameters at 15% addition compared to 5%. The thickness of the control remained the same at 5% addition regardless of the storage time, but either of the starches reduced the thickness at longer storage time, significantly (Table 2). Nonetheless, Osan starch increased the thickness, significantly, compared to the control or the native starch especially after 0 or 30-day storage (Table 2). The same trend was observed for the 15% addition, where the thickness decreased after longer storage time for all samples, but the drop was less for Osan starch (Table 3). The firmness of all three patties increased after longer storage time regardless of the added fat or starch level. The increase in the firmness after longer storage time can be attributed to the moisture loss, however, Osan starch exhibited the least increase in firmness. Yang et al. [42] reported the addition of 4% modified waxy maize starch to low-fat frankfurters leading to reduced moisture loss by up to 7.2%. However, Claus and Hunt [43] also reported that, modified waxy maize starch applied to low-fat bologna was more effective in controlling moisture losses relative to native wheat starch, which is consistent with a very low retrogradation of waxy starch [34,35]. There are conflicting reports regarding the effect of starch on the cooking loss of meat products, especially for low muscle products such as frankfurters and bologna [33]. Cooking loss increase was observed for all samples at longer storage time, but Osan cooking loss was much less than the other treatments (Table 2 and Table 3).

#### 3.3.3. Sensory Attributes of Cooked Beef Meat Patties

The sensory evaluation results are presented in Table 4. The incorporation of native or Osan starches at 5% did not have any significant effect on the juiciness, flavor, color, or acceptability of the patties, but the tenderness was improved significantly (*p* ≤ 0.05). The addition of 15% of either starches significantly improved the juiciness, flavor and, the overall acceptability of the patties, whereas the tenderness and the color were slightly improved more with Osan starch. After storage, the sensory evaluation showed superior performance of Osan over the control and the native starch, whereas longer storage appeared to have negative effect on the parameter of samples containing native starch (Table 5). Once again, samples with 15% Osan starch scored higher than those with 5%. Nonetheless, storage at −20 °C for 60 days appeared to have limited effect on the overall acceptability of the patties, but Osan starch patties scored higher. Higher starch content facilitates stable protein-starch matrices, where hydrogen and covalent bonding and charge-charge interactions occur [44].

#### 3.3.4. Color of Raw and Cooked Patties

Consumer acceptability of meat products is dependent on its color because it is indicative of freshness. The effect of starch on the surface color of beef patties is presented in Table 6. The control sample exhibited the lowest redness (a*) values compared to the native and Osan starches, regardless of the added amount, but Osan starch had higher a* value. The incorporation of plant-based material was reported to increase the a* value of cooked meat [45]. Higher a* values of Osan starch-containing patties compared to control and the native starch indicate color stabilization, since the reduction in a* values is suggestive of myoglobin oxidation and the formation of net myoglobin [46]. Other researchers reported increase in meat redness after incorporation of potato starches, whereas cassava starch reduced the a*. Therefore, the type of starch as well as the amount of the added starch can be considered as factors that affect beef patties redness. This is obvious on the magnitude of the effect of native or Osan starch on the a* of the patties (Table 6). The amount of the incorporated Osan (5 or 15%) had slight effect on a* (Table 6). The redness of the control increased as a function of the added amount, but it increased significantly after storage for 60 days (Table 7). Sample with Osan had the highest a* as a function of storage time (60 days). The lightness (l*) of the samples and the control stayed almost the same at 5% incorporation, but at 15% Osan starch exhibited significantly higher l* value. The higher lightness could be attributed to the dilution of the color by the added starch. Reports in the literature mentioned reduction or stability of the l* based on the incorporated material into the patties [47]. No reduction in lightness was observed after storage for 30 or 60 days for either 5 or 15% incorporation. The yellowness (b*) of the control increased compared to the starch-containing samples where Osan exhibited the most b* value, but after storage for 60 days a drop in b* values was observed (Table 7).

The color of the cooked patties 6 h after preparation is presented in Table 8. The redness (a*) of cooked patties showed no significant difference between the control at 5% addition and Osan, but Osan starch exhibited significantly higher a* at 15% addition, whereas native starch reduced the redness. The lightness (l*) and yellowness (b*) of cooked patties was reduced in the presence of native or Osan starches. The effect of storage time on the color of the cooked patties is listed in Table 9. The redness (a*) of the cooked control did not change significantly after storage at either of the fat additions, whereas Osan starch increased the redness of the cooked patties compared to the control or native starch through storage time, especially after 60 days. Samples exhibited an increase in lightness and yellowness after longer storage time regardless of the amount of the added starch or animal fat (Table 9). The ratio of a*/b* is often used to describe color quality. The redness of the control was significantly reduced by the addition of 15% animal fat compared to 5%, but native starch maintained similar color for both additions. The addition of 5% Osan starch significantly reduced the yellowness which is obvious on the a*/b* value (Table 6). Therefore, animal fat added at 15% and Osan starch at 5% had the most improvement on the color of the cooked patties, which could be interpreted as stabilizing effect of Osan starch. This led to the higher redness, which indicates myoglobin color stability. Reports in the literature showed how the addition of processed plant leaf materials can reduce the a*/b* of beef patties which means lower redness [48].

## 4. Conclusions

In this study we examined the quality parameters of patties prepared from ground beef and processed with the addition of Osan starch consequently revealing varying positive changes in the quality properties of the final product. The addition of Osan starch did not alter the pH level or the organoleptic standards of the product. The most notable change was manifested by the increase in textural and microstructural properties and the significant improvement of the cooking characteristics such as: yield, moisture retention, patties redness, thickness, and decrease in cooking loss. Scanning electron microscope (SEM) images of the samples confirmed the rise of intermolecular interaction between the proteins and the Osan starch, which resulted in small pores on the surface of the patties.

## Figures and Tables

**Figure 1 foods-10-01157-f001:**
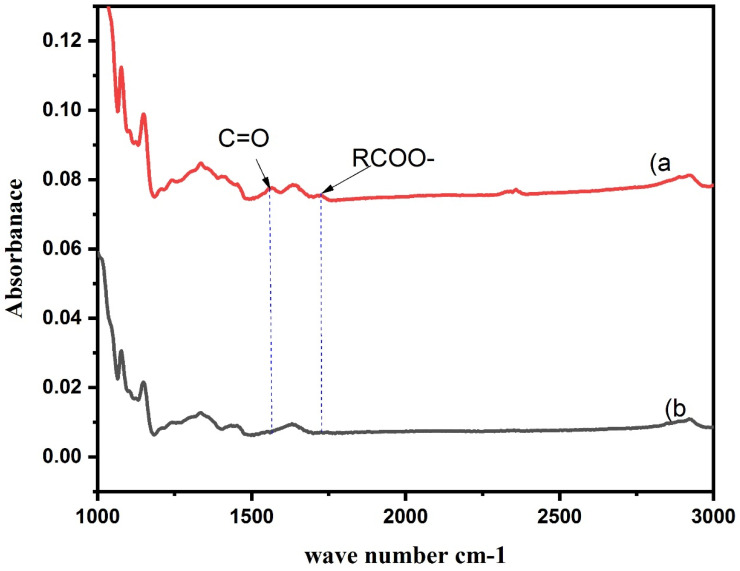
FT-IR spectra of (**a**) Osan-corn starch and (**b**) native corn starch.

**Figure 2 foods-10-01157-f002:**
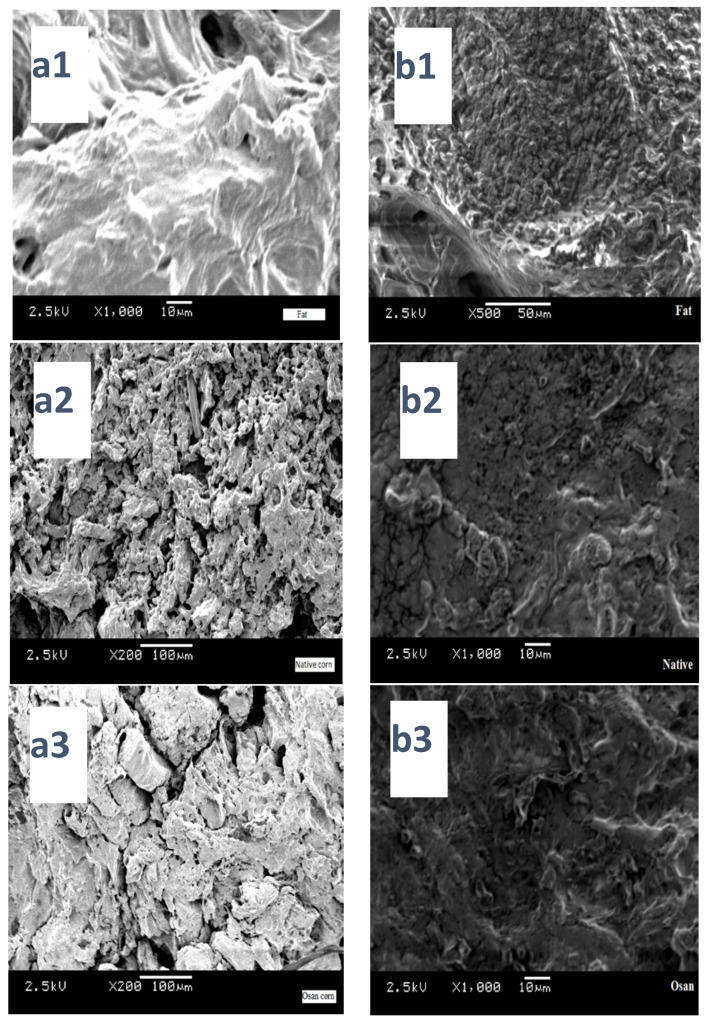
Scanning electron microscope (SEM) images of cooked patties containing animal fat, native corn starch and Osan corn starch (**a1**) control, (**a2**) native starch, and (**a3**) Osam starch 15% addition; (**b1**) control, (**b2**) native starch, and (**b3**) Osan 5% addition.

**Table 1 foods-10-01157-t001:** The effect of 5 and 15% native, octenyl-succinylated (Osan) corn starch, and 5% animal fat on the physiochemical characteristics of beef patties.

	%					
	Control ^7^		Native Starch		Osan Starch	
	5%	15%	5%	15%	5%	15%
M.C.U ^2^	69.81 ± 0.72 ^1^	63.72 ± 0.25 ^b,1^	70.04 ± 0.12 ^a^	66.25 ± 0.40 ^a^	70.47 ± 0.31 ^a^	64.41 ± 0.50 ^b^
M.C.C ^3^	42.45 ± 0.15 ^b^	51.19 ± 0.10 ^b^	60.69 ± 0.20 ^a^	56.56 ± 0.83 ^a^	61.14 ± 0.34 ^a^	56.90 ± 0.32 ^a^
C. loss	41.22 ± 0.18 ^a^	45.29 ± 0.50 ^a^	19.53 ± 0.11 ^b^	09.50 ± 0.38 ^b^	18.27 ± 0.20 ^c^	08.45 ± 0.58 ^b^
M.R ^4^	35.82 ± 0.53 ^c^	43.83 ± 0.44 ^b^	69.96 ± 0.58 ^b^	77.37 ± 1.31 ^a^	71.20 ± 0.58 ^a^	79.50 ± 0.80 ^a^
Diameter ^5^	18.37 ± 0.27 ^a^	23.45 ± 1.08 ^a^	13.36 ± 1.01 ^b^	16.44 ± 2.05 ^b^	07.71 ± 0.38 ^c^	08.63 ± 2.51 ^c^
D.S ^6^	24.02 ± 0.19 ^a^	28.98 ± 0.14 ^a^	17.87 ± 0.26 ^b^	18.02 ± 0.07 ^b^	15.86 ± 0.13 ^c^	10.81 ± 0.16 ^c^
Thickness ^8^	06.32 ± 0.49 ^c^	14.51 ± 0.20 ^b^	14.02 ± 1.57 ^b^	07.09 ± 0.95 ^c^	25.05 ± 0.28 ^a^	18.10 ± 0.00 ^a^
Firmness	121.97 ± 7.72 ^a^	88.89 ± 4.25 ^a^	43.56 ± 1.19 ^b^	78.51 ± 2.08 ^b^	40.96 ± 1.70 ^b^	71.62 ± 1.54 ^c^
pH	5.55 ± 0.01 ^b^	5.53 ± 0.00 ^a^	5.66 ± 0.06 ^a^	5.60 ± 0.02 ^b^	5.69 ± 0.02 ^a^	5.62 ± 0.03 ^a^

^1^ The statistical analysis was done separately for the 5% and for the 15%; ^2^ M.C.U = moisture content of uncooked patties. ^3^ M.C.C = moisture content of cooked patties. ^4^ M.R = moisture retention; ^5^ diameter reduction; ^6^ D.S = dimensional shrinkage. ^7^ control = animal fat; ^8^ thickness increase, ^a–c^ Values followed by different letters within each row are significantly different (*p* ≤ 0.05).

**Table 2 foods-10-01157-t002:** Effect of 5% octenyl-succinylated (Osan) corn starch, native starch, and animal fat on the physiochemical characteristics of meat patties after storage at −20 °C for 0, 30, and 60 days.

		Days		
Parameter		0	30	60
Cooking loss	Control ^6^	41.22 ± 0.18 ^c,1^	51.85 ± 0.50 ^b^	57.80 ± 0.62 ^a^
	Native	19.53 ± 0.11 ^c^	28.17 ± 1.35 ^b^	36.10 ± 0.14 ^a^
	Osan	18.27 ± 0.20 ^c^	27.39 ± 0.23 ^b^	33.58 ± 1.33 ^a^
Diameter ^2^	Control	18.37 ± 0.27 ^b^	20.71 ± 0.58 ^a^	21.70 ± 0.54 ^a^
	Native	13.36 ± 1.01 ^b^	15.62 ± 0.04 ^a^	16.66 ± 0.14 ^a^
	Osan	07.71 ± 0.38 ^b^	09.99 ± 1.66 ^a,b^	11.75 ± 0.75 ^a^
D. S ^3^	Control	24.02 ± 0.19 ^c^	26.08 ± 0.08 ^b^	27.79 ± 0.21 ^a^
	Native	17.87 ± 0.26 ^b^	21.25 ± 0.16 ^a^	22.36 ± 1.07 ^a^
	Osan	15.86 ± 0.13 ^b^	16.38 ± 0.38 ^a,b^	16.54 ± 0.48 ^a^
Thickness ^4^	Control	06.32 ± 0.49 ^a^	05.80 ± 0.45 ^a^	05.61 ± 0.98 ^a^
	Native	14.02 ± 1.57 ^a^	08.25 ± 0.07 ^b^	06.13 ± 0.47 ^b^
	Osan	25.05 ± 0.28 ^a^	12.20 ± 0.67 ^b^	07.47 ± 0.00 ^c^
Firmness ^5^	Control	121.97 ± 7.72 ^c^	209.57 ± 4.05 ^b^	307.19 ± 0.81 ^a^
	Native	43.56 ± 1.19 ^c^	92.93 ± 7.40 ^b^	127.84 ± 8.59 ^a^
	Osan	40.96 ± 1.70 ^c^	82.73 ± 0.05 ^b^	112.61 ± 3.84 ^a^
pH	Control	05.63 ± 0.07 ^a^	05.39 ± 0.05 ^b^	05.35 ± 0.02 ^b^
	Native	5.66 ± 0.06 ^a^	5.39 ± 0.00 ^b^	5.34 ± 0.02 ^b^
	Osan	05.69 ± 0.02 ^a^	05.33 ± 0.02 ^b^	05.36 ± 0.01 ^b^

^1^ Values followed by different letters (^a–c^) within each row are significantly different (*p* ≤ 0.05); ^2^ diameter reduction; ^3^ D.S = dimensional shrinkage; ^4^ thickness increase; ^5^ firmness in Newton; ^6^ control = animal fat.

**Table 3 foods-10-01157-t003:** Effect of 15% octenyl-succinylated (Osan) corn starch, native starch, and animal fat on the physiochemical characteristics of meat patties after storage at −20 °C for 0, 30, and 60 days.

		Days		
Parameter		0	30	60
Cooking loss	Control ^6^	45.29 ± 0.50 ^b,1^	53.43 ± 1.65 ^a^	55.08 ± 0.67 ^a^
	Native	09.50 ± 0.38 ^c^	20.45 ± 1.97 ^b^	23.67 ± 1.32 ^a^
	Osan	08.45 ± 0.58 ^c^	17.86 ± 0.15 ^b^	21.15 ± 1.39 ^a^
Diameter ^2^	Control	22.63 ± 0.18 ^a^	23.23 ± 1.33 ^a^	24.51 ± 0.31 ^a^
	Native	14.03 ± 0.13 ^c^	16.63 ± 0.04 ^b^	18.68 ± 0.71 ^a^
	Osan	05.43 ± 0.30 ^b^	09.59 ± 0.06 ^a^	10.87 ± 0.97 ^a^
D. S ^3^	Control	28.98 ± 0.14 ^a^	29.93 ± 1.96 ^a^	30.60 ± 0.30 ^a^
	Native	18.02 ± 0.07 ^b^	20.85 ± 1.22 ^a^	22.53 ± 0.99 ^a^
	Osan	10.81 ± 0.16 ^b^	14.84 ± 0.50 ^a^	15.33 ± 1.10 ^a^
Thickness ^4^	Control	14.51 ± 0.20 ^a^	07.45 ± 0.70 ^b^	02.72 ± 0.00 ^c^
	Native	07.09 ± 0.95 ^a^	05.21 ± 0.09 ^a,b^	04.12 ± 0.45 ^b^
	Osan	18.10 ± 0.00 ^a^	14.39 ± 0.60 ^b^	10.18 ± 0.00 ^c^
Firmness ^5^	Control	88.89 ± 4.25 ^c^	163.63 ± 9.22 ^b^	278.68 ± 14.65 ^a^
	Native	78.50 ± 2.08 ^b^	137.84 ± 6.30 ^a^	158.23 ± 16.32 ^a^
	Osan	71.62 ± 1.54 ^c^	100.37 ± 8.98 ^b^	127.81 ± 8.63 ^a^
pH	Control	05.53 ± 0.00 ^a^	05.43 ± 0.00 ^b^	05.19 ± 0.00 ^c^
	Native	05.60 ± 0.02 ^a^	05.32 ± 0.00 ^b^	05.07 ± 0.01 ^c^
	Osan	05.58 ± 0.04 ^a^	05.38 ± 0.05 ^b^	05.12 ± 0.05 ^c^

^1^ Values followed by different letters (^a–c^) within each row are significantly different (*p* ≤ 0.05); ^2^ diameter reduction; ^3^ D.S = dimensional shrinkage; ^4^ thickness increase; ^5^ firmness in Newton; ^6^ control = animal fat.

**Table 4 foods-10-01157-t004:** The effect of different levels of octenyl-succinylated (Osan) corn starch, native starch, and animal fat on the sensory characteristics of meat patties. The test was based on a 9 points hedonic.

Sensory Characteristics	Control ^2^		Native Starch		Osan Starch	
	5%	15%	5%	15%	5%	15%
Tenderness	7.30 ± 0.26 ^a,b,1^	6.43 ± 1.08 ^b,1^	7.53 ± 1.12 ^b^	7.53 ± 1.05 ^a^	8.30 ± 0.65 ^a^	8.03 ± 0.87 ^a^
Juiciness	6.53 ± 1.26 ^a^	5.96 ± 0.82 ^c^	7.07 ± 1.18 ^a^	6.80 ± 1.21 ^b^	7.53 ± 1.26 ^a^	8.23 ± 0.83 ^a^
Flavor	7.23 ± 1.36 ^a^	6.61 ± 1.19 ^b^	7.53 ± 1.19 ^a^	7.23 ± 1.23 ^a,b^	7.84 ± 0.98 ^a^	7.89 ± 0.95 ^a^
Color	7.92 ± 0.95 ^a^	7.48 ± 1.12 ^a^	7.38 ± 1.26 ^a^	7.59 ± 1.02 ^a^	7.84 ± 0.80 ^a^	7.92 ± 0.93 ^a^
Acceptability	6.42 ± 1.22 ^a^	6.50 ± 1.13 ^b^	7.50 ± 1.32 ^a^	7.61 ± 1.24 ^a^	7.61 ± 1.30 ^a^	7.78 ± 1.08 ^a^

^1^ The statistical analysis was done separately for the 5% and for the 15%; ^(a–c)^ values followed by different letters within each row and addition percent are significantly different (*p* ≤ 0.05); ^2^ control = animal fat; the statistical analysis was done for the 5%, separate from the 15%.

**Table 5 foods-10-01157-t005:** Effects of different levels of octenyl succinylated (Osan) corn starch, native starch, and animal fat on the sensory characteristics of meat patties after storage at −20 for different days.

		Days					
		0		30		60	
Parameter		5%	15%	5%	15%	5%	15%
Tenderness	Control ^1^	7.30 ± 0.94 ^a,1^	^1^ 6.43 ± 1.08 ^a^	6.30 ± 1.31 ^b^	6.38 ± 1.19 ^a^	6.53 ± 0.96 ^a,b^	6.46 ± 0.96 ^a^
	Native	7.53 ± 1.12 ^a^	7.53 ± 1.05 ^a,b^	7.46 ± 1.26 ^a^	7.84 ± 0.98 ^a^	7.15 ± 1.28 ^a^	6.65 ± 1.12 ^b^
	Osan	8.38 ± 0.65 ^a^	8.03 ± 0.87 ^a^	7.92 ± 1.18 ^a,b^	8.07 ± 1.03 ^a^	7.46 ± 0.87 ^b^	7.84 ± 1.28 ^a^
Juiciness	Control	6.38 ± 1.12 ^a^	5.96 ± 0.82 ^a^	6.38 ± 1.04 ^a^	6.34 ± 1.06 ^a^	6.46 ± 0.96 ^a^	5.92 ± 0.86 ^a^
	Native	7.07 ± 1.18 ^a^	6.80 ± 1.2 ^b^	7.84 ± 0.84 ^a^	8.00 ± 0.81 ^a^	6.92 ± 1.25 ^a^	6.94 ± 1.19 ^a,b^
	Osan	7.53 ± 1.26 ^a^	8.23 ± 0.83 ^a^	7.53 ± 1.05 ^a^	7.53 ± 1.05 ^a^	7.53 ± 0.66 ^a^	7.38 ± 0.96 ^a^
Flavor	Control	7.46 ± 1.12 ^a^	6.61 ± 1.19 ^a^	7.00 ± 1.29 ^a^	6.92 ± 1.03 ^a^	6.69 ± 1.10 ^a^	9.92 ± 1.18 ^a^
	Native	7.53 ± 1.19 ^a^	7.23 ± 1.23 ^a^	8.16 ± 0.83 ^a^	7.53 ± 0.96 ^a^	7.53 ± 0.96 ^a^	6.92 ± 1.32 ^a^
	Osan	7.84 ± 0.98 ^a^	7.89 ± 0.95 ^a^	7.00 ± 1.15 ^a^	7.46 ± 1.05 ^a^	7.07 ± 1.32 ^a^	7.00 ± 0.91 ^a^
Color	Control	7.92 ± 0.95 ^a^	7.48 ± 1.12 ^a^	7.23 ± 1.36 ^a^	7.23 ± 1.16 ^a^	7.15 ± 1.28 ^a^	7.46 ± 1.05 ^a^
	Native	7.38 ± 1.26 ^a^	7.59 ± 1.02 ^a^	7.84 ± 1.14 ^a^	7.92 ± 0.95 ^a^	7.69 ± 1.18 ^a^	7.34 ± 1.14 ^a^
	Osan	7.84 ± 0.80 ^a^	7.92 ± 0.93 ^a^	7.69 ± 0.94 ^a^	8.23 ± 0.72 ^a^	7.92 ± 1.11 ^a^	7.42 ± 1.18 ^a^
Acceptability	Control	6.77 ± 0.95 ^a^	6.50 ± 1.13 ^a^	6.65 ± 1.23 ^a^	6.95 ± 1.09 ^a^	6.76 ± 1.05 ^a^	6.81 ± 1.06 ^a^
	Native	7.50 ± 1.32 ^a^	7.61 ± 1.24 ^a^	7.64 ± 0.96 ^a^	7.73 ± 1.06 ^a^	7.12 ± 1.20 ^a^	7.18 ± 0.89 ^a^
	Osan	7.73 ± 1.11 ^a^	7.78 ± 1.08 ^a^	7.69 ± 1.02 ^a^	7.63 ± 1.20 ^a^	7.45 ± 0.81 ^a^	7.20 ± 1.04 ^a^

^1^ Control = animal fat; (^a–c^) values followed by different letters within each row and addition percent are significantly different (*p* ≤ 0.05); the statistical analysis was done for the 5% separate from the 15%.

**Table 6 foods-10-01157-t006:** The effect of octenyl-succinylated (Osan) corn starch, native starch, and animal fat on the color characteristics of un-cooked meat patties after 6 h.

	Control ^2^		Native Starch		Osan Starch	
	5%	15%	5%	15%	5%	15%
L* ^3^	36.96 ± 1.68 ^a,1^	40.08 ± 0.32 ^a,1^	33.17 ± 0.36 ^b^	33.00 ± 0.71 ^c^	32.47 ± 0.44 ^b^	37.59 ± 0.31 ^b^
a* ^4^	3.43 ± 0.46 ^a^	2.46 ± 0.19 ^b^	2.68 ± 0.51 ^b^	2.80 ± 0.16 ^b^	3.62 ± 0.08 ^a^	3.49 ± 0.10 ^a^
b* ^5^	14.40 ± 0.51 ^a^	15.66 ± 0.07 ^a^	10.55 ± 0.37 ^b^	13.36 ± 0.12 ^c^	10.18 ± 0.21 ^b^	14.86 ± 0.23 ^b^
a*/b*	0.24 ± 0.41	0.16 ± 0.08	0.25 ± 0.21	0.21 ± 0.51	0.36 ± 0.31	0.23 ± 0.11

^1^ Values followed by different letters (^a–c^) within each row are significantly different (*p* ≤ 0.05); ^2^ control = animal fat. ^3^ L* = lightness, ^4^ a* = redness and ^5^ b* = yellowness; the statistical analysis was done for the 5% separate from the 15%.

**Table 7 foods-10-01157-t007:** The effect of octenyl-succinylated (Osan) corn starch, native starch, and animal fat on the color characteristics of un-cooked meat patties at −20 for different days.

		0		30		60	
		5%	15%	5%	15%	5%	15%
L* ^3^	Control ^2^	^1^ 36.96 ± 1.68 ^a^	^1^ 40.08 ± 0.32 ^a^	36.84 ± 0.52 ^a^	34.76 ± 0.69 ^c^	38.26 ± 0.30 ^a^	36.29 ± 0.54 ^b^
	Native	33.17 ± 0.36 ^b^	33.00 ± 0.71 ^b^	35.24 ± 0.20 ^a^	36.85 ± 0.85 ^a^	35.30 ± 0.52 ^a^	37.18 ± 0.10 ^a^
	Osan	32.47 ± 0.44 ^b^	37.59 ± 0.31 ^b^	37.82 ± 1.27 ^a^	39.40 ± 0.69 ^a^	38.14 ± 0.40 ^a^	38.95 ± 0.56 ^a,b^
a* ^4^	Control	3.43 ± 0.18 ^a^	2.46 ± 0.19 ^b^	3.00 ± 0.18 ^a^	3.28 ± 0.04 ^a^	3.20 ± 0.19 ^a^	3.39 ± 0.32 ^a^
	Native	2.68 ± 0.46 ^b^	2.80 ± 0.16 ^b^	3.72 ± 0.33 ^b^	4.88 ± 0.10 ^a^	5.50 ± 0.45 ^a^	4.49 ± 0.30 ^a^
	Osan	3.62 ± 0.08 ^b^	3.49 ± 0.10 ^c^	3.72 ± 0.33 ^b^	4.89 ± 0.30 ^b^	4.62 ± 0.12 ^a^	6.80 ± 0.77 ^a^
b* ^5^	Control	14.40 ± 0.51 ^a^	15.66 ± 0.07 ^a^	12.81 ± 0.89 ^a^	13.21 ± 0.49 ^b^	15.68 ± 0.16 ^a^	13.57 ± 1.51 ^b^
	Native	10.55 ± 0.37 ^c^	13.36 ± 0.12 ^b^	14.67 ± 0.25 ^b^	14.90 ± 0.89 ^a^	15.68 ± 0.09 ^a^	16.15 ± 0.15 ^a^
	Osan	10.18 ± 0.21 ^c^	14.86 ± 0.23 ^b^	14.67 ± 0.59 ^b^	15.19 ± 0.57 ^b^	15.69 ± 0.59 ^a^	16.70 ± 0.15 ^a^

^1^ Values followed by different letters (^a–c^) within each row are significantly different (*p* ≤ 0.05); ^2^ control = animal fat. ^3^ L* = lightness, ^4^ a* = redness and ^5^ b* = yellowness; the statistical analysis was done for the 5% separate from the 15%.

**Table 8 foods-10-01157-t008:** The effect of octenyl-succinylated (Osan) corn starch, native starch, and animal fat on the color characteristics of cooked meat patties after 6 h.

	^2^ Control		Native Starch		Osan Starch	
	5%	15%	5%	15%	5%	15%
L* ^3^	^1^ 31.46 ± 1.68 ^a^	^1^ 30.46 ± 0.32 ^a^	32.17 ± 0.36 ^b^	32.38 ± 0.71 ^c^	31.49 ± 0.44 ^b^	38.50 ± 0.31 ^b^
a* ^4^	9.52 ± 0.46 ^a^	9.69 ± 0.19 ^b^	10.30 ± 0.51 ^b^	10.91 ± 0.16 ^b^	11.42 ± 0.08 ^a^	11.66 ± 0.10 ^a^
b* ^5^	8.72 ± 0.51 ^a^	8.32 ± 0.07 ^a^	9.65 ± 0.37 ^b^	11.77 ± 0.12 ^c^	9.96 ± 0.21 ^b^	11.88 ± 0.23 ^b^
a*/b*	1.09 ± 0.41	1.16 ± 0.08	1.01 ± 0.21	0.93 ± 0.51	1.15 ± 0.31	0.98 ± 0.11

^1^ Values followed by different letters (^a–c^) within each row are significantly different (*p* ≤ 0.05); ^2^ control = animal fat. ^3^ L* = lightness, ^4^ a* = redness and ^5^ b* = yellowness; the statistical analysis was done for the 5% separate from the 15%.

**Table 9 foods-10-01157-t009:** The effect of octenyl-succinylated (Osan) corn starch, native starch and animal fat on the color characteristics of cooked meat patties at −20 for different days.

		0		30		60	
		5%	15%	5%	15%	5%	15%
L* ^3^	Control ^2^	31.46 ± 1.68 ^b,1^	^1^ 30.46 ± 0.32 ^b^	32.45 ± 0.52 ^b^	36.47 ± 0.69 ^a^	34.89 ± 0.30 ^a^	37.27 ± 0.54 ^a^
	Native	32.86 ± 0.36 ^b^	32.38 ± 0.71 ^c^	40.82 ± 0.20 ^a^	42.64 ± 0.85 ^b^	36.41 ± 0.52 ^b^	45.63 ± 0.10 ^a^
	Osan	31.49 ± 0.44 ^b^	38.50 ± 0.31 ^b^	36.96 ± 1.27 ^a^	45.52 ± 0.69 ^a^	38.09 ± 0.40 ^a^	46.43 ± 0.56 ^a^
a* ^4^	Control	8.27 ± 0.18 ^a^	9.69 ± 0.19 ^a^	8.79 ± 0.18 ^a^	10.02 ± 0.04 ^a^	9.06 ± 0.19 ^a^	8.28 ± 0.32 ^b^
	Native	9.65 ± 0.46 ^b^	10.91 ± 0.16 ^a^	13.38 ± 0.33 ^a^	9.83 ± 0.10 ^a^	12.22 ± 0.45 ^a^	9.90 ± 0.30 ^a,b^
	Osan	9.96 ± 0.08 ^b^	11.66 ± 0.10 ^a^	11.75 ± 0.33 ^a,b^	11.80 ± 0.30 ^a^	12.65 ± 0.12 ^a^	10.67 ± 0.77 ^a^
b* ^5^	Control	9.52 ± 0.51 ^a^	8.32 ± 0.07 ^b^	8.64 ± 0.89 ^a^	10.72 ± 0.49 ^a^	7.91 ± 0.16 ^a^	10.52 ± 1.51 ^a^
	Native	10.30 ± 0.37 ^a,b^	11.77 ± 0.12 ^c^	9.83 ± 0.25 ^b^	14.42 ± 0.89 ^b^	10.76 ± 0.09 ^a^	15.48 ± 0.15 ^a^
	Osan	11.24 ± 0.21 ^a^	11.88 ± 0.23 ^b^	10.17 ± 0.59 ^b^	15.40 ± 0.57 ^a^	10.78 ± 0.59 ^a,b^	15.43 ± 0.15 ^a^

^1^ Values followed by different letters (^a–c^) within each row are significantly different (*p* ≤ 0.05); ^2^ control = animal fat. ^3^ L* = lightness, ^4^ a* = redness and ^5^ b* = yellowness; the statistical analysis was done for the 5% separate from the 15%.

## Data Availability

Data will be available upon request from the corresponding author.

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
