# Peer review of "Quality Characteristics of Beef Patties Prepared with Octenyl-Succinylated (Osan) Starch"

_foods, 2021, doi:10.3390/foods10061157_

Round 1

Reviewer 1 Report

Title:

 Quality Characteristics of Beef Patties Prepared with 1 Octenyl-Succinylated (Osan) Starch

The authors investigated some characteristics beef patties obtained replacing fat with  Octenyl-Succinylated (Osan) Starch

The manuscript have serious concerns that should be considered.

The organization of the paper must be revised completely, section and subsection have to be a logical sequence (for example the subsection 2.2.4  Scanning electron microscopy (SEM) of the patties  is before subsection 2.2.5 2.2.5. Preparation and processing of beef patties)

The author do not follow the rule of the format required from Foods

The introduction is poor with few references and unclear

The aim of the study is unclear

Materials and methods are incorrectly and poorly. Sensory analysis are conducted incorrectly

Octenyl-Succinylated (Osan) Starch have to be declared if it is foodgrade

Author Response

Response to Reviewer 1

At first, my colleague and I would like to thank you for taking the time to comment on our work.

Quality Characteristics of Beef Patties Prepared with Octenyl-Succinylated (Osan) Starch

The authors investigated some characteristics beef patties obtained replacing fat with Octenyl-Succinylated (Osan) Starch

The manuscript has serious concerns that should be considered.

  1. The organization of the paper must be revised completely, section and subsection have to be a logical sequence (for example the subsection 2.2.4.  Scanning electron microscopy (SEM) of the patties  is before subsection 2.2.5 2.2.5. Preparation and processing of beef patties)

Response: The sections were in the correct order in accordance to the methods section  

The author doesn’t follow the rule of the format required from Foods

Response: The paper was formatted according to the Food rules.

The introduction is poor with few references and unclear

Response: The introduction included 16 paper of current related literature, which is sufficient in our opinion when compared to other publication. By the way, all the papers mentioned are published between the years 2006 and 2021.

The aim of the study is unclear

Response: The objectives of this work were compare the quality characteristics of beef patties prepared with starch to those with animal fat.  Therefore, ocetinyle succenylated and native corn starch as fat replacement in beef patties was utilized. This work includes the effect of storability at -20⁰C on the quality characteristics of the patties (microstructure, cooking properties, moisture retention, texture, color and sensory characteristics of the produced patties).

Materials and methods are incorrectly and poorly. Sensory analysis is conducted incorrectly

Response: The sensory evaluation was clarified by adding the following sentence “The sensory test was performed using a 9 points hedonic test (Affective Tests), which includes scale from 1 (dislike extremely) to 9 (like extremely) in a single session. This test is useful for evaluating the acceptance of new products”.     

Octenyl-Succinylated (Osan) Starch have to be declared if it is food grade

Response:  Based on the literature and the amount of Osan was added, this product is considered food grade based on two references we sited on section 2.2.1. Octenyl-Succinylation of corn starch (Osan).

My opinion is that the manuscript is not suitable for publication in foods.

Reviewer 2 Report

In this work the authors propose the use of Octenyl-Succinylated (Osan) modified Starch in ground beef patties. The results are interested and prove the advantage of modified starches in food industry. The manuscript suffers from a moderate presentation that underestimates the value of the work.

Specific points

Abstract

L 12 'as controlled was used' instead of 'and the control was considered'

L 16 - Shrinking cannot be an improvement, please describe correctly

L 17 - Change 'on' to 'of'

L 24 - others types are not acceptable? please report scores of all types of starches

Introduction

L 52 - Change including to 'improving'

L 62 - Change 'its' to 'their'

L 63 change 'chemical modifications' to 'chemically modified starches'

L 76 reference by Partheniadis et al Pharmaceutics 2020, 12, 571; doi:10.3390/pharmaceutics12060571 may be a suitable addition 

L 86 - change replacer to replacement

Materials and methods

L 104 -change modified to sythesized

L 107 - change NAoH to NAOH

L 108 Here should be given the chemical name e.g. octenyl succinic anhydride of the reactant

L 177 change 'were' to 'was'

Results and discussion

Figure 1. Shift spectra vertically and focus on the region of interest to improve demonstration of CO and RCOO peaks

Figure 2. Do not leave empty space laterally between images

 Line 344 - add missin 'e' after don

Table 1 and other Tables - superscripts a, b, c are not explained in the footnote

Author Response

Response to Reviewer 1

At first, my colleague and I would like to thank you for taking the time to comment on our work.

Specific points

The response was done on the paper in red color.

Abstract

L 12 'as controlled was used' instead of 'and the control was considered'

Response: Was done

L 16 - Shrinking cannot be an improvement, please describe correctly

Response: Along with Osan, native corn starch was used as control and considered the patties with added animal fat. The data showed that Osan reduced the cooking loss and dimensional shrinkage

L 17 - Change 'on' to 'of'

Response: was done

L 24 - others types are not acceptable? please report scores of all types of starches

Response: The overall sensory acceptability scores were given to cooked patties containing Osan starch as well as the native starch, whereas as 15% animal fat was favored too.

Introduction

L 52 - Change including to 'improving'

Response: was done

L 62 - Change 'its' to 'their'

Response: was done

L 63 change 'chemical modifications' to 'chemically modified starches'

Response: was done

L 76 reference by Partheniadis et al Pharmaceutics 2020, 12, 571;

doi:10.3390/pharmaceutics12060571 may be a suitable addition 

Response: reference was added “The use of sodium octenyl succinate starch in a methacrylate/Polysaccharides blends, introduced good flow ability, surface-active, smoother particle surface and low-viscosity to spray dried emulsion of the blends [14].”

L 86 - change replacer to replacement

Response: was done

Materials and methods

L 104 -change modified to synthesized

Response: was done

L 107 - change NAoH to NAOH

Response: was done

L 108 Here should be given the chemical name e.g. octenyl succinic anhydride of the reactant

Response: was done

L 177 change 'were' to 'was'

Response: was done

Results and discussion

Figure 1. Shift spectra vertically and focus on the region of interest to improve demonstration of

CO and RCOO peaks

Response:

Figure 2. Do not leave empty space laterally between images

Response: Sorry, the software did not allow me to erase the space

Line 344 - add missin 'e' after don

Response: was done

Table 1 and other Tables - superscripts a, b, c are not explained in the footnote

Response: Was done for all Tables.

Round 2

Reviewer 1 Report

The paper is remarkably improved. The author have to improve the references following the instructions of authors

Author Response

The paper is remarkably improved. The author have to improve the references following the instructions of authors

Response: references were modified according to Foods